# Effects of Nitrite Exposure on the Hematological Properties, Antioxidant and Stress Responses of Juvenile Hybrid Groupers, *Epinephelus lanceolatus* ♂ × *Epinephelus fuscoguttatus* ♀

**DOI:** 10.3390/antiox11030545

**Published:** 2022-03-14

**Authors:** Jun-Hwan Kim, Yue Jai Kang, Kyung Mi Lee

**Affiliations:** 1Department of Aquatic Life and Medical Science, Sun Moon University, Asan 31460, Korea; kyj5088@hanmail.net or; 2National Institute of Fisheries Science, West Sea Fisheries Research Institute, Incheon 22383, Korea; bioykm@korea.kr

**Keywords:** hybrid grouper, nitrite exposure, hematological properties, antioxidant response, stress indicator

## Abstract

Nitrite concentrations can reach high levels in indoor aquaculture systems, thus it is vital to determine the nitrite tolerance of aquaculture fish species. Here, juvenile hybrid groupers (*Epinephelus lanceolatus* ♂ × *Epinephelus fuscoguttatus* ♀, Family: Serranidae) were exposed to waterborne nitrite at 0, 10, 20, 40, and 80 mg NO_2_^−^/L for 2 weeks. Nitrite exposure caused significant reductions in hematocrit and hemoglobin levels, significant increases in plasma calcium and plasma ALP levels, but had no significant effects on magnesium and total protein levels. Of the antioxidant responses investigated, SOD activity increased significantly in the liver and gills, but GST activity and GSH levels were significantly inhibited by nitrite exposure. Stress indicators, such as plasma cortisol and HSP 70 levels, were significantly stimulated by nitrite exposure. In brief, nitrite exposure over 20 mg NO_2_^−^/L had toxic effects and affected the hematological properties, antioxidant responses, and stress indicators of juvenile hybrid groupers.

## 1. Introduction

Nitrite is a critical toxic substance for organisms in the aquatic environment that is generated in the process of bacterial nitrification of ammonia or the denitrification of nitrate [1]. Generally, nitrite presently occurs at low concentrations in the aquatic environments, but can significantly increase in high-density aquaculture or eco-friendly aquaculture systems, such as Recirculating Aquaculture Systems (RAS) and bio-floc technology (BFT) [2,3]. Imbalance in nitrification activity of bacteria, such as Nitrosomonas and Nitrobacter species, results in nitrite accumulation in the aquaculture environment [4]. Exposure to elevated levels of nitrite exposure in the aquatic environments leads to the accumulation of nitrite in the body of aquatic animals, which can have adverse toxic effects [5].

Nitrite toxicity in fish is caused by the competitive inhibition of chloride uptake, which reduces the extra- and intra-cellular chloride levels [6]. Nitrite in the body of fish affects the exchange of Cl^−^/HCO_3_^−^ ions in the chloride-secreting cells of the gill tissue. Since nitrite ions compete with chloride ions, fish in environments with low salt concentrations are more susceptible to nitrite exposure [7]. Nitrite exposure affects the osmo-regulatory functions, ionic homeostasis, and metabolism in various fish species [8,9]. In addition, the increase in nitrite disturbs various physiological functions, such as growth, nitrogen excretion, ion regulation, and can cause reproduction retardation, endocrine disruption, respiratory problems, and ultimately increased mortality rates [10].

Upon exposure, nitrite from the aquatic environment accumulates in the circulatory system as well as in tissues, such as those of the gill, liver, spleen, muscle, and brain, and affects the fish blood physiology [11]. Nitrite anions can accumulate in the plasma via the circulating blood from the branchial epithelial cells [12]. Nitrite exposure to fish directly affects their hematological properties of fish by oxidizing hemoglobin and affects the cardiovascular functions [13]. The mechanism for cellular nitrite absorption is not yet been precisely defined, but nitrite is known to enter red blood cells (RBCs) primarily by diffusion of nitrous acid (HNO_2_) through the lipid bilayer, conductive transport channels, or nitrite anion (NO_2_^−^) diffusion via the RBC anion exchange [12]. The extracellular nitrite entering the RBC membrane reacts with active deoxyhemoglobin, and reduces NO by oxidizing ferrous heme to ferric heme, resulting in the formation of dysfunctional methemoglobin [14]. Exposure to high nitrite concentration induces an increase in methemoglobin and disturbs physiological functions in fish species [15]. Hematological parameters are a sensitive and reliable indicator of the health status of fish. Considering that nitrite exposure causes an increase in methemoglobin and affect blood properties, which are important indicators for evaluating physiological health, hematological properties can be important indicators for determining the toxic effects of nitrite exposure on fish.

Nitrite exposure generates reactive oxygen species (ROS) and adversely affects the DNA damage in fish via nitrite toxicity, by oxidative damage involving DNA, proteins, and lipids with inflammatory response stimulation [16]. Antioxidant defense systems in aquatic animals are effective mechanisms to deal with the oxidative stress induced by nitrite exposure [17]. Of the various antioxidant enzyme responses, superoxide dismutase (SOD) is the first defense mechanism against oxidative stress, and SOD activity stimulates the change from superoxide anions to oxygen (O_2_) and hydrogen peroxide (H_2_O_2_) [18]. Glutathione (GSH) systems play a critical role in controlling the redox state of the cells, and GSH and GSH-S-transferases (GST) are major components of these systems [19]. GSH plays a key role in controlling the activity of antioxidant enzymes against oxidative stress induced by nitrite exposure, and in enhancing the growth performance [20]. GST plays a role in free radical removal by oxidizing GSH into GSSG to protect cells from oxidative injury [21]. Exposure to nitrite in aquatic environment can lead to oxidative stress induced by excessive ROS in fish, and confirmation of the antioxidant reaction should be a major indicator of oxidative stress from nitrite exposure.

Exposure to high concentrations of nitrite can induce stress in aquatic animals due to physiological changes and tissue damage [22]. Considering that nitrite exposure acts as an environmental stressor for fish inducing a direct stress response to the toxic exposure, stress indicators can be sensitive indicators of the toxic effects of nitrite exposure [5]. Of the many stress indicators used, cortisol is a general corticosteroid that is used to evaluate quantitative stress in teleosts from environmental stressors, because cortisol receptors control proper immune function, physiological metabolism, and homeostasis [23]. In addition, cortisol is a sensitive indicator used to distinguish between stressed and normal states with a marked response to acute stress [24]. Heat shock proteins (HSPs) in fish are major stress-associated proteins that activate various stress factors, and these proteins are reliable indicators for evaluating the stress status in fish exposed to nitrite [11]. HSP 70 functions as a molecular chaperone to control protein homeostasis, preventing aggregation, and the refolding of misfolded proteins, and is a reliable indicator of fish stress from nitrite exposure [16,25].

The hybrid grouper (*Epinephelus lanceolatus* ♂ × *Epinephelus fuscoguttatus* ♀, Family: Serranidae) is a hybrid fish species created by the Golden Seed Project for the development of high-quality seed. It is a fast-growing, disease-resistant, and high-water-temperature (25–35 °C)-tolerant fish species. Because fish losses are greatly attributed to high temperatures in summer, the high-water-temperature resistance of the hybrid grouper makes it a potentially excellent alternative aquaculture fish species. In contrast to the high-water-temperature tolerance of this species, the hybrid grouper is vulnerable to low water temperatures; therefore, it is necessary to breed hybrid groupers in indoor aquaculture systems, such as RAS and BFT, in the winter season. However, nitrite concentrations can reach high levels in RAS and BFT aquaculture systems. Although nitrite tolerance in hybrid groupers is essential, investigations of this aspect are currently limited. This study aimed to determine the physiological resistance limit of the hybrid grouper to nitrite exposure by investigating certain blood properties, antioxidant levels, and stress responses following nitrite exposure. Such knowledge can be used to guide future hybrid grouper breeding standards.

## 2. Materials and Methods

### 2.1. Pre-Acute Nitrite Exposure Experiment

Hybrid grouper (*Epinephelus fuscoguttatus* ♀ × *E. lanceolatus* ♂) (mean weight: 36.3 ± 3.9 g, mean length: 13.0 ± 0.5 cm) were exposed to different nitrite concentrations at 0, 100, 200, 400, 800, and 1600 mg NO_2_^−^/L. The mortality at 1600 mg NO_2_^−^/L increased rapidly, resulting in 100% mortality. The mortality at 800 mg NO_2_^−^/L at 96 h was 33.3%. The lethal concentration for 50% (LC_50_) of hybrid grouper for nitrite exposure was 856.79 mg NO_2_^−^/L, and the 96 h LC_50_ of hybrid grouper is demonstrated in Table 1 [26].

### 2.2. Experimental Fish

Juvenile hybrid grouper, *E. lanceolatus* ♂ × *E. fuscoguttatus* ♀ (mean weight: 36.8 ± 4.2 g, mean length: 13.1 ± 0.6 cm) were obtained from a local fish farm in Korea. Fish used in this experiment were kept in a lab environment for 3 weeks with feeding twice a day for fish health status before the experiment, and the seawater components are as shown in the Table 2. Nitrite exposure was made in a standard stock solution of 40,000 mg NO_2_^−^/L using NaNO_2_ (Sigma Chemical, St. Louis, MO, USA), followed by concentrations of 0, 10, 20, 40, and 80 mg NO_2_^−^/L in each 150 L circular tank. A number of 60 fish (6 fish/tank × 5 concentrations × 2 periods) were used to evaluate nitrite toxic effects on fish. The water tank for each concentration was completely changed once every 2 days, and then prepared using a standard stock solution. The actual nitrite concentration was measured using a nitrite analysis kit (Merck & Co., Inc., Kenilworth, NJ, USA) as shown in Table 3. At the 1 and 2 weeks, the blood and tissues were collected after sufficient anesthesia for 30 s at 5 ppm using MS-222 (Sigma Chemical, St. Louis, MO, USA).

### 2.3. Blood Physiology

Blood was collected using a disposable syringe treated with heparin in the tail vein of the fish. Hematological parameters such as hematocrit and hemoglobin were measured immediately after blood sampling. Hematocrit value was centrifuged after placing blood in the capillary tube to confirm the value through a hematocrit measuring plate. Hemoglobin concentration was determined using the Cyan-methemoglobin technique (Asan Pharm. Co., Ltd., Seoul, Korea). After analyzing the blood properties, blood was separated from the plasma by centrifugation at 10,000 rpm for 5 min at 4 °C. The plasma components such as calcium, magnesium, total protein, and ALP (alkaline phosphatase) were measured using an Asan clinical kit according to the methods of Kim et al. [27].

### 2.4. Antioxidant Responses

Liver and gill tissues were collected to assess antioxidant responses. The sampled liver and gill tissues were homogenized by diluting 10-fold using 0.1 M PBS buffer immediately after collection. The supernatant was collected by centrifugation at 10,000 rpm for 30 min at 4 °C, and used for antioxidant analysis. The SOD (superoxide dismutase) and GST (glutathione S-transferase) activities and GSH (glutathione) level were analyzed by the methods of Kim et al. [28].

### 2.5. Stress Indicators

Plasma cortisol concentration was measured with a monoclonal antibody enzyme-linked immunosorbent assay (ELISA) quantification kit (Enzo Life Sciences, Inc., Farmingdale, NY, USA). Liver and gill tissues were homogenized by diluting 10-fold using 0.1 M PBS buffer immediately after collection. The supernatant was collected by centrifugation at 10,000 rpm for 30 min at 4 °C, and used for HSP 70 (Heat shock protein 70). HSP 70 in the liver and gills was measured using the ELISA assay kit (MyBioSource, Inc., San Diego, CA, USA). The plasma cortisol and HSP 70 were measured by the methods of Kim et al. [27].

### 2.6. Statistical Analysis

We performed statistical analysis using the SPSS/PC+ statistical package (SPSS Inc., Chicago, IL, USA) based on the results of this study. Multiple comparisons evaluated the significance between groups using the Tukey’s test using one-way analysis of variance (ANOVA), with a significance level of *p* < 0.05.

### 2.7. Ethics Approval and Consent to Participate

This study was conducted with the research ethics approval of the Institutional Animal Care and Use Committee of the National Institute of Fisheries Science (2019-NIFS-IACUC-29). In addition, all researchers have completed animal protection, animal welfare, and animal experimentation conducted by the National Institute of Fisheries Science.

## 3. Results

### 3.1. Blood Physiology

Hematological properties of hybrid grouper, *E. lanceolatus* ♂ × *E. fuscoguttatus* ♀ exposed to waterborne nitrite are shown in Figure 1. Hematocrit values was significantly decreased in the 80 mg NO_2_^−^/L at 1 week and over 40 mg NO_2_^−^/L at 2 weeks. Hemoglobin concentration was significantly decreased over 20 mg NO_2_^−^/L at 1 week and over 10 mg NO_2_^−^/L at 2 weeks.

Plasma calcium was significantly decreased in the 80 mg NO_2_^−^/L both at 1 and 2 weeks. However, there was no significant change in the plasma magnesium and total protein. Plasma ALP was significantly increased over 40 mg NO_2_^−^/L both at 1 and 2 weeks.

### 3.2. Antioxidant Responses

Antioxidant responses of hybrid grouper, *E. lanceolatus* ♂ × *E. fuscoguttatus* ♀ exposed to waterborne nitrite are shown in Figure 2. The liver SOD activity was significantly increased over 40 mg NO_2_^−^/L at 1 week and over 20 mg NO_2_^−^/L at 2 weeks. A significant increase in the gill SOD activity was observed over the 10 mg NO_2_^−^/L both at 1 and 2 weeks.

The liver GST activity was significantly decreased in the 80 mg NO_2_^−^/L at 2 weeks, whereas no significant change was observed at 1 week. The gill GST activity was significantly decreased over 40 mg NO_2_^−^/L at 1 week and over 10 mg NO_2_^−^/L at 2 weeks.

The liver GSH level was significantly decreased over 10 mg NO_2_^−^/L both at 1 and 2 weeks. In the gill GSH level, a significant decrease was observed in the 80 mg NO_2_^−^/L at 1 week and in the 20 and 80 mg NO_2_^−^/L at 2 weeks.

### 3.3. Stress Responses

Plasma cortisol of hybrid grouper, *E. lanceolatus* ♂ × *E. fuscoguttatus* ♀ exposed to waterborne nitrite are shown in Figure 3. The plasma cortisol was significantly increased over 20 mg NO_2_^−^/L at 1 week. At 2 weeks, the plasma cortisol was also significantly over 20 mg NO_2_^−^/L.

HSP 70 in the liver and gills of hybrid grouper, *E. lanceolatus* ♂ × *E. fuscoguttatus* ♀ exposed to waterborne nitrite are shown in Figure 4. In the liver HSP 70, a significant stimulation was observed over 40 mg NO_2_^−^/L at 1 week and over 10 mg NO_2_^−^/L at 2 weeks. The gill HSP 70 was significantly stimulated over 40 mg NO_2_^−^/L at 1 week and over 20 mg NO_2_^−^/L at 2 weeks.

## 4. Discussion

Nitrite exposure in aquatic animals directly affects their blood physiology by altering their blood acid–base homeostasis (Madison and Wang, 2006). In addition, nitrite exposure decreases the oxygen affinity of hemoglobin as a result of the conversion of Fe^2+^ to Fe^3+^ in the heme group [29], which results in hypoxia. Methemoglobin in fish exposed to nitrite is formed by oxygenated RBC and methemoglobin reductase in the hepatocytes [30]. Nitrite exposure was shown to induce a significant decrease in the hematocrit and hemoglobin levels of Amazonian fish, *Colossoma macropomum*, because of RBC shrinkage, hemolysis, and RBC life-cycle reduction [31]. These authors suggested that the methemoglobin reductase system is generally activated to convert methemoglobin to hemoglobin under nitrite exposure stress, and the resulting high metabolic energy requirement affects the life span of RBCs. Avilez et al. [32] reported a significant decrease in the hematocrit and hemoglobin levels of neotropical teleost matrinxã, *Brycon cephalus*, exposed to environmental nitrite, and suggested that these decreases were due to blood cell lysis. Yildiz et al. [33] also reported significant decrease in the hematocrit and hemoglobin levels of Nile tilapia, *Oreochromis niloticus*, by nitrite exposure. In this study, hematological parameters, such as hematocrit and hemoglobin levels, were significantly decreased in the hybrid grouper, *E. lanceolatus* ♂ × *E. fuscoguttatus* ♀. These significant reductions indicated that nitrite exposure caused structural damage to the erythrocytes, thus resulting in hemolytic anemia.

Nitrite exposure affects plasma components because it actively enters plasma as nitrite ions with chloride ions via the Cl^−^/HCO_3_^−^ exchange mechanism of the branchial epithelium [29]. Nitrite exposure in fish can affect the electrolyte balance in the plasma, and serveral studies have demonstrated a significant change in the ionic composition of plasma of fish exposed to nitrite [34]. Among inorganic plasma components, plasma calcium and magnesium are critical indicators of ion regulation and homeostasis, as well as constant cellular maintenance and enzymatic activity [35]. Kim et al. [36] reported significant alterations in the plasma calcium and magnesium levels of juvenile olive flounders, *Paralichthys olivaceus*, by waterborne zinc exposure. In this study, the plasma calcium level of the hybrid grouper was significantly decreased, whereas no change was detected in the plasma magnesium level, which indicated that nitrite exposure impaired ion regulation in this species.

Plasma protein is an important plasma component and a reliable indicator of fish health status [27]. A significant decrease in plasma protein levels in various fish species following nitrite exposure has been reported, which suggests that protein depletion is induced by protein leakage into peripheral fluids, in addition to protein proteolysis by nitrite exposure stress [11,37,38]. However, in this study, nitrite exposure did not cause a significant change in the total protein levels in plasma of the hybrid grouper.

Plasma alkaline phosphatase (ALP) is considered to be a critical indicator of tissue damage in fish exposed to various environmental stressors, and this activity in fish is influenced by nitrite exposure. Jia et al. [39] reported that nitrite exposure caused a significant increase in the plasma ALP of juvenile turbot, *Scophthalmus maximus*, as a result of hepatic necrosis induced by the toxic effects of nitrite exposure. The plasma ALP level of the hybrid grouper was significantly increased by nitrite exposure, and this significant increase may have been induced by tissue injury. Nitrite exposure led to blood toxicity, and had various effects on the hematological properties and plasma components of juvenile hybrid groupers.

Nitrite exposure causes oxidative stress in aquatic animals, and can increase the oxygen free radicals in the body, which induce free radical accumulation and result in multiple oxidative stress-induced toxic effects in animals [40]. Teleosts have developed antioxidant defense systems to protect the tissues from oxidative damage caused by the overproduction of free radicals in fish [41]. SOD is a fundamental component of the early defense system and antioxidant response to oxidative stress due to nitrite exposure, and Kim et al. [42] reported a significant increase in the SOD activity of *P. olivaceus* exposed to nitrite. Guo et al. [43] also reported a significant increase in the SOD activity of red swamp crayfish, *Procambarus clarkii*, by nitrite exposure, and suggested that the increase in SOD activity was due to the activation of the antioxidant system to prevent oxidative stress by ROS overproduction. In the present study, the SOD activity of the hybrid grouper was significantly increased by nitrite exposure, and this significant increase was presumedly indicative of an increase in enzyme activity to remove excess free radicals induced by stress exposure.

GST is a critical enzyme that is involved in liver detoxification, and it excretes endogenous or exogenous compounds by catalyzing the oxidation of GSH to GSSG [44]. Kim and Kang [45] suggested that GST is a major bio-indicator of oxidative stress in fish under environmental stress due to its ability to eliminate and prevent the accumulation of toxicants in the cellular system. Mohamed et al. [46] reported a significant decrease in the GST activity of Nile tilapia, *O. niloticus*, after hexavalent chromium exposure, and suggested that a decrease in GST is induced by the stimulation of the biotransformation process. In previous studies, oxidative stress caused by toxic exposure to various environmental stresses increased the GST activity in various fish species [3,27,36]; however, in the present study, the observed decrease in GST activity is thought to have been due to exhaustion from the excessive removal of free radicals. In other words, it is judged that the decrease in GST is due to a functional loss on account of the generation of active oxygen in excess of the ability of the antioxidant enzyme.

GSH is a major antioxidant response in cells, and a stable balance between GSH and GSSG is essential to maintain physiological homeostasis and cell function [47]. Lin et al. [20] reported a significant decrease in the GSH level of bighead carp, *Aristichthys nobilis,* by nitrite exposure. These authors suggested that GSH depletion was caused by the oxidation of GSH to GSSH to protect the cells from free radicals, which indicates an activation of the antioxidant function in response to nitrite exposure. Kim et al. [3] also reported a significant decrease in the GSH level of olive flounders, *P. olivaceus,* exposed to nitrite. Lin et al. [48] also reported a significant decrease in the GSH level of zebrafish, Danio rerio, exposed to nitrite, for the purpose of ROS elimination. In the present study, the GSH levels of the hybrid grouper were significantly depleted by nitrite exposure, which is presumedly due to the oxidation of GSH to reduce ROS accumulation and oxidative stress induced by nitrite exposure. Given that changes in antioxidant responses are considered reliable biomarkers to evaluate oxidative stress in fish exposed to various environmental stressors [49], changes in the antioxidant responses of the hybrid grouper in this study indicate that nitrite exposure induced oxidative stress in this species.

Environmental toxicity causes physiological changes and oxidative stress and acts as a stress factor for fish [50]. Of the various stress indicators, cortisol is a sensitive indicator of toxic effects in fish exposed to environmental stressors. Gao et al. [1] reported a significant increase in the cortisol of puffer fish, *Takifugu rubripes*, exposed to nitrite, and they suggested that the stimulation was caused by the activation of the hypothalamic–pituitary–interrenal axis in response to stress induced by nitrite exposure. Jia et al. [15] also suggested that nitrite exposure caused a significant increase in the cortisol of juvenile turbot, *S. maximus*. Consistently, we previously observed a significant increase in the cortisol of olive flounders, *P. olivaceus*, by nitrite exposure, which was induced by the stress-induced stimulation in lymphopenia of the lymphoid tissues [5]. In the present study, the plasma cortisol of the hybrid grouper was also significantly stimulated by nitrite exposure, which is presumedly a result of the stress response induced by nitrite exposure.

HSPs have major functions in protecting newly synthesized protein and proper protein folding, and HSP 70 is the most conserved HSP that has a critical function in protecting the cells of fish under oxidative stress conditions [51]. Gao et al. [11] reported a significant increase in the HSP 70 of pufferfish, *T. rubripes*, exposed to nitrite, which may represent a protective mechanism to prevent protein damage and misfolding of proteins. Jia et al. [39] also reported that nitrite exposure induced a significant increase in the HSP 70 of juvenile turbot, *S. maximus*. In the present study, nitrite exposure caused a significant increase in the HSP 70 of the hybrid grouper, which indicates nitrite-induced stress and protein damage caused by nitrite exposure. On the other hand, since HSP 70 participates in antioxidant reactions to catalyze the conversion of free radicals [52], a significant increase in HSP 70 in this study may indicate an association with an antioxidant reaction.

## 5. Conclusions

In conclusion, the results of this study indicated that waterborne nitrite exposure negatively impacted certain hematological parameters, such as hematocrit and hemoglobin levels, in juvenile hybrid groupers and altered the plasma components, such as plasma calcium and ALP levels. Oxidative stress from nitrite exposure also caused a significant increase in SOD activity and a decrease in GST activity and GSH levels in the hybrid grouper. The levels of stress indicators, i.e., cortisol and HSP 70, were significantly increased by nitrite exposure. Collectively, the results of this study suggested that subacute exposure to waterborne nitrite at levels higher than 20 mg NO_2_^−^/L caused physiological changes in juvenile hybrid groupers due to the toxic effects.

## Figures and Tables

**Figure 1 antioxidants-11-00545-f001:**
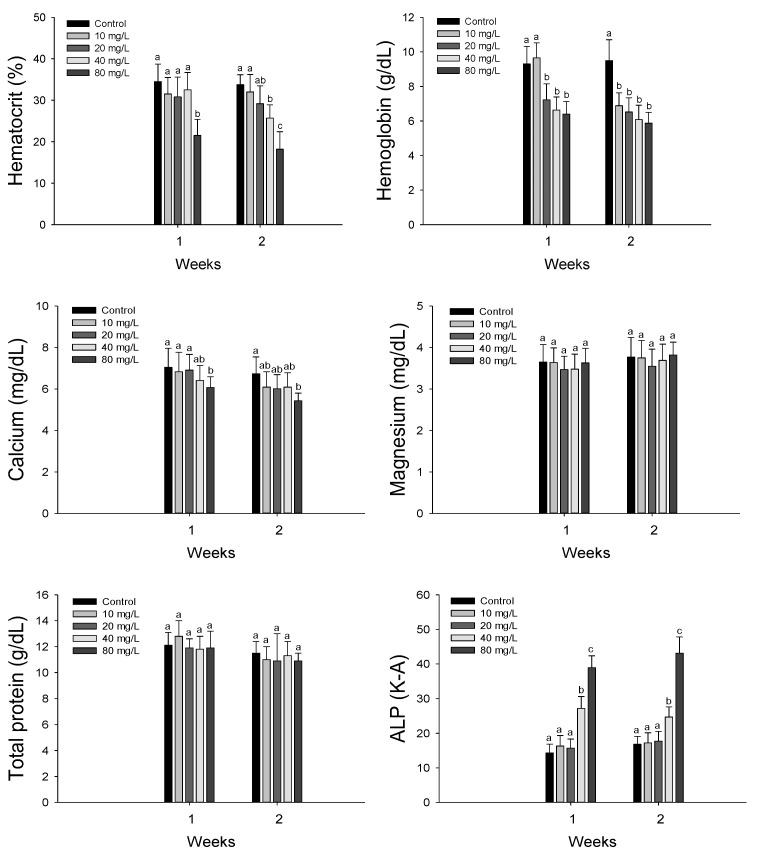
Changes of blood physiology in hybrid groupers, *Epinephelus lanceolatus* ♂ × *Epinephelus fuscoguttatus* ♀ exposed to waterborne nitrite for 2 weeks. Vertical bar denotes a standard error. Values with different superscript are significantly different (*p* < 0.05) as determined by the Tukey’s multiple range test.

**Figure 2 antioxidants-11-00545-f002:**
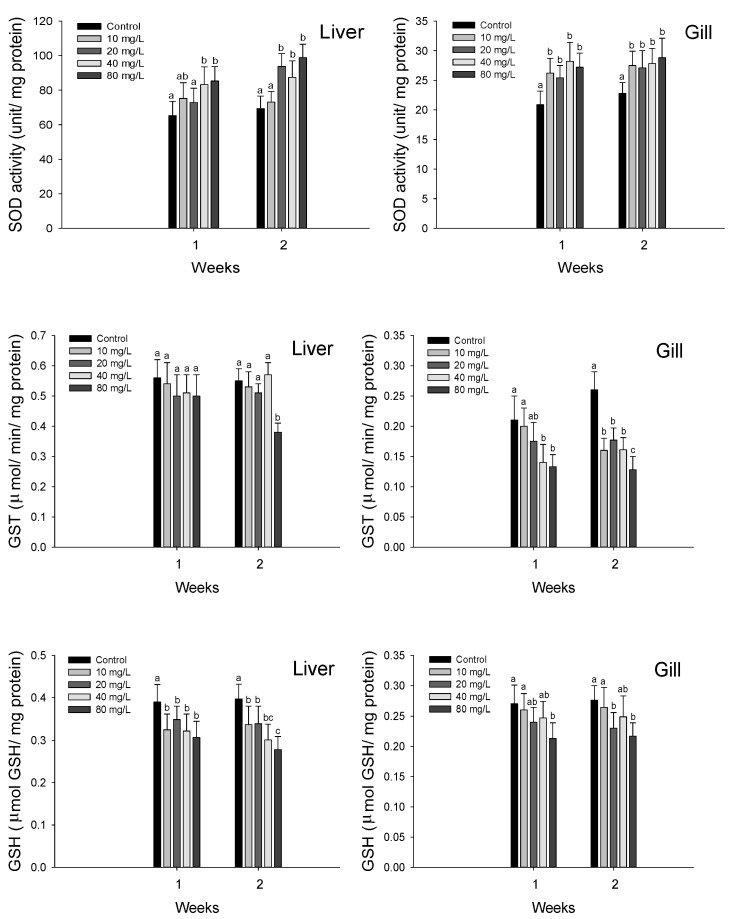
Changes of antioxidant responses in hybrid groupers, *Epinephelus lanceolatus* ♂ × *Epinephelus fuscoguttatus* ♀ exposed to waterborne nitrite for 2 weeks. Vertical bar denotes a standard error. Values with different superscript are significantly different (*p* < 0.05) as determined by the Tukey’s multiple range test.

**Figure 3 antioxidants-11-00545-f003:**
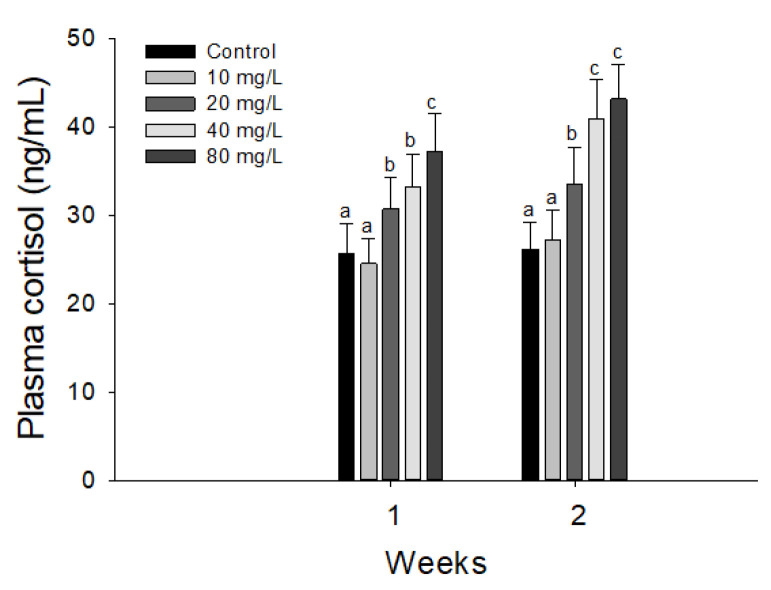
Changes of plasma cortisol in hybrid groupers, *Epinephelus lanceolatus* ♂ × *Epinephelus fuscoguttatus* ♀ exposed to waterborne nitrite for 2 weeks. Vertical bar denotes a standard error. Values with different superscript are significantly different (*p* < 0.05) as determined by the Tukey’s multiple range test.

**Figure 4 antioxidants-11-00545-f004:**
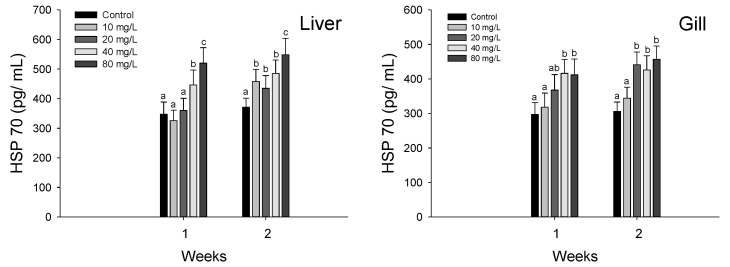
Changes of HSP 70 in hybrid groupers, *Epinephelus lanceolatus* ♂ × *Epinephelus fuscoguttatus* ♀ exposed to waterborne nitrite for 2 weeks. Vertical bar denotes a standard error. Values with different superscript are significantly different (*p* < 0.05) as determined by the Tukey’s multiple range test.

**Table 1 antioxidants-11-00545-t001:** Lethal concentration (LC_50_) of hybrid grouper (*Epinephelus fuscoguttatus* ♀ × *E. lanceolatus* ♂) exposed to the different concentration of waterborne nitrite for 96 h.

95% Confidence Limits	
Probability	Estimate (mg/L)
0.01	538.39
0.10	681.39
0.20	741.60
0.30	785.02
0.40	822.11
0.50	856.79
0.60	891.46
0.70	928.56
0.80	971.97
0.90	1032.19
0.99	1175.18

**Table 2 antioxidants-11-00545-t002:** The chemical components of seawater and experimental condition used in the experiments.

Item	Value (Mean ± SD)
Temperature (°C)	24.8 ± 0.5
pH	8.38 ± 0.11
Salinity (‰)	31.9 ± 0.3
Dissolved Oxygen (mg/L)	7.48 ± 0.24
Ammonia (mg/L)	0.58 ± 0.13
Nitrite (mg/L)	0.06 ± 0.02
Nitrate (mg/L)	1.12 ± 0.15

**Table 3 antioxidants-11-00545-t003:** Analyzed waterborne nitrite concentration (NO_2_^−^ mg/L) from each source.

Waterborne Nitrite Concentration (NO_2_^−^ mg/L)					
Waterborne nitrite levels	0	10	20	40	80
Measured nitrite levels	0.07	10.5	21.2	43.9	83.3

## Data Availability

Data is contained within the article.

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
