# Peer review of "Effects of Nitrite Exposure on the Hematological Properties, Antioxidant and Stress Responses of Juvenile Hybrid Groupers, Epinephelus lanceolatus ♂ × Epinephelus fuscoguttatus"

_antioxidants, 2022, doi:10.3390/antiox11030545_

Round 1

Reviewer 1 Report

I think it is an interesting work on a oxidative stress responses of an fish species of aquacultural interest. The objective of the work is rightly justified and the methodology suits it. Although the results were expectable, authors have demonstrated them in a suitable way.

Therefore I think only a few minor concerns should be corrected before publication (see attach).

Author Response

  1. You use this abbreviation 9 times but it is not explained in either of them. Please indicate the abbreviation the first time you use it in Abstract and body text.

: As the reviewer said, explanations for abbreviations have been added to the Materials and Methods section.

2.3. Blood physiology

The plasma components such as calcium, magnesium, total protein, and ALP (alkaline phosphatase) were measured using Asan clinical kit according to the methods of Kim et al. (2020b).

2.4. Antioxidant responses

The SOD (superoxide dismutase) and GST (glutathione S-transferase) activities and GSH (glutathione) level were analyzed by the methods of Kim et al. (2017a).

2.5. Stress indicators

HSP 70 (Heat shock protein 70) in the liver and gills was measured using the ELISA assay kit (MyBioSource, Inc., USA). The plasma cortisol and HSP 70 were measured by the methods of Kim et al. (2020b).

  1. We revise the wrong expression as below.

2.1. Pre-acute nitrite exposure experiment

The lethal concentration for 50% (LC50) of hybrid grouper for nitrite exposure was 856.79 mg NO2-/L, and the 96h LC50 of hybrid grouper was demonstrated in Table 1.

2.2. Experimental fish

Fish used in this experiment were kept in a lab environment for 3 weeks before the experiment, and the seawater components are as shown in the Table 1. 

At the 1 and 2 weeks, the blood and tissues were collected after sufficient anesthesia for 30 seconds at 5 ppm using MS-222 (Sigma Chemical, St. Louis, MO, USA).

  1. Please indicate mean and dispersion parameter ....SD?, SE?

: We revise the Table 1 from Value to Value (mean ± SD) as your suggestion.

  1. Please indicate briefly the sample homogenization/extraction process.

: We add more content for the sample homogenization/extraction process as below.

2.5. Stress indicators

Plasma cortisol concentration was measured with a monoclonal antibody enzyme-linked immunosorbent assay (ELISA) quantification kit (Enzo Life Sciences, Inc., Farmingdale, NY, USA). Liver and gill tissues were homogenized by diluting 10-fold using 0.1 M PBS buffer immediately after collection. The supernatant was collected by centrifugation at 10,000 rpm for 30 min at 4°C, and used for HSP 70 (Heat shock protein 70). HSP 70 in the liver and gills was measured using the ELISA assay kit (MyBioSource, Inc., USA). The plasma cortisol and HSP 70 were measured by the methods of Kim et al. (2020b).

  1. Please indicate abbreviations: ALP (K-A). All figures should be self-explanatory

: We add the full name in the Materials and Methods as your suggestion.

2.3. Blood physiology

The plasma components such as calcium, magnesium, total protein, and ALP (alkaline phosphatase) were measured using Asan clinical kit according to the methods of Kim et al. (2020b).

  1. µmol

: We revise the figures as your suggestion.

  1. As above, please explain abbreviations.

: We add the full name in the Materials and Methods as your suggestion.

2.4. Antioxidant responses

The SOD (superoxide dismutase) and GST (glutathione S-transferase) activities and GSH (glutathione) level were analyzed by the methods of Kim et al. (2017a).

2.5. Stress indicators

HSP 70 (Heat shock protein 70) in the liver and gills was measured using the ELISA assay kit (MyBioSource, Inc., USA). The plasma cortisol and HSP 70 were measured by the methods of Kim et al. (2020b).

  1. Please "refresh" this abbreviation: indicate again Red Blood Cells

: We revise the RBCs to Red Blood Cells as your comment.

In addition, nitrite exposure decreases the oxygen affinity of hemoglobin as a result of the conversion of Fe2+ to Fe3+ in the heme group (Jensen, 2003), which results in hypoxia. Methemoglobin in fish exposed to nitrite is formed by oxygenated Red Blood Cells and methemoglobin reductase in hepatocytes (Madison and Wang, 2006). 

  1. Refresh abbreviation (not explained yet)

: We add the full name of ALP in the discussion as your suggestion.

Plasma ALP (alkaline phosphatase) is considered to be a critical indicator of tissue damage in fish exposed to various environmental stressors, and the plasma ALP activity in fish is influenced by nitrite exposure. 

  1. Is there any work supporting it? Please explain it.

: We support the paragraph as your suggestion

In previous studies, oxidative stress caused by toxic exposure to various environmental stresses increased the GST activity in various fish species (Kim et al., 2019b; Kim et al., 2020a, b); however, in the present study, the observed decrease in GST activity is thought to have been due to exhaustion from the excessive removal of free radicals. In other words, it is judged that the decrease in GST is due to the functional loss due to the generation of active oxygen in excess of the ability of the antioxidant enzyme.

We sincerely thanks for your careful review to improve this article.

Reviewer 2 Report

The manuscript is about the evaluation of nitrite toxic to hybrid grouper via tracking the LC50, hematological parameters and antioxidant. However, the manuscript needs an English editing by a native English speaker. Therefore, I suggest this manuscript needs a major revision before publication.

  1. In the materials and methods, 2.1 section seems like results, it should be move to results section, and a method of LC50 is needed.
  2. Data shown in Table 1 should rearranged into a figure for easier to understand.
  3. In subsection 2.2, line 4, it should be table 2 rather than table 1.
  4. table 2 is proper to be move to supplementary.
  5. Please check Table 3, the unite format should be corrected, and the word size should be consistent.
  6. All methods of physiological parameters should be detailed and not just give a citation.
  7. Please recheck all data shown in this study, as you can see that many data shown in figures 1-4 among groups are overlapping, but they are marked by different letters means significantly different. It is odd.

Author Response

The manuscript is about the evaluation of nitrite toxic to hybrid grouper via tracking the LC50, hematological parameters and antioxidant. However, the manuscript needs an English editing by a native English speaker. Therefore, I suggest this manuscript needs a major revision before publication.

: This manuscript has already been translated into English by a native English speaker. As the reviewer said, we entrusted the English proofreading to a better manuscript. However, since the due date for re-submission is 10 days, first, we will entrust the English translation and reflect it in the manuscript after submitting the revised version.

In the materials and methods, 2.1 section seems like results, it should be move to results section, and a method of LC50 is needed.

: Part 2.1 is the acute experimental part that corresponds to the preceding research prior to conducting this study. So putting it in the results doesn't seem appropriate. Instead, references to LC50 following acute nitrite exposure are included as below.

The lethal concentration for 50% (LC50) of hybrid grouper for nitrite exposure was 856.79 mg NO2-/L, and the 96h LC50 of hybrid grouper was demonstrated in Table 1 (Cho et al., 2020).

Cho, J.H., Kim, S.R., Hur, Y.B., Lee, K.M., Kim, J.H., 2020. Tolerance limit of nitrite exposure to hybrid grouper (Epinephelus fuscoguttatus♀נE. lanceolatus♂): hematological parameters and plasma components.Korean Journal of Environmental Biology 38, 93-100.

In subsection 2.2, line 4, it should be table 2 rather than table 1.

: We revise the wrong written sentence as your suggestion.

2.2. Experimental fish

Fish used in this experiment were kept in a lab environment for 3 weeks with feeding twice a day for fish health status before the experiment, and the seawater components are as shown in the Table 2. 

table 2 is proper to be move to supplementary.

: We ask again if moving Table 2 to Supplement means to remove it from the manuscript. We will follow the opinions of the reviewer.

Please check Table 3, the unite format should be corrected, and the word size should be consistent.

: We revise the wrong font size as your comment.

Table 3. Analyzed waterborne nitrite concentration (NO2- mg/L) from each source.

Waterborne nitrite concentration (NO2- mg/L)

Waterborne nitrite levels

0

10

20

40

80

Measured nitrite levels

0.07

10.5

21.2

43.9

83.3

All methods of physiological parameters should be detailed and not just give a citation.

: When the contents were described in detail, the similarity of the papers was high because of the same analysis technique, so only references were inevitably included. If it is possible to indicate the same analysis technique described in the previous paper, we will include it.

Please recheck all data shown in this study, as you can see that many data shown in figures 1-4 among groups are overlapping, but they are marked by different letters means significantly different. It is odd.

: As the reviewer said, we feel strange in the figure. There is nothing wrong with the picture in word, but there is a part where the picture becomes strange when it is put in the format of the antioxidants journal. We will inquire about this part with the person in charge of the journal.

We sincerely thanks for your careful review to improve this article.

Reviewer 3 Report

This manuscript evaluated effects of nitrite exposure on the hematological properties, antioxidant and stress response of juvenile hybrid groupers. The research is meaningful. However, results of the experiment were a bit poor. The mechanism of response to nitrite exposure needs to investigate in depth.

  1. L117: NO2-/L should be NO2-/L.
  2. L143: table 1 should be table 2.
  3. Please provide feeding information before experiment.
  4. L149: How to ensure the stable nitrite concentration of each tank when change water once every 2 days?
  5. There were 10 tanks for 5 concentrations and 2 periods, thus for one treatment only one tank? No replications?
  6. Please check font size in table 3.
  7. L181: 10,000 rpm was not too high?
  8. The experiment was designed 2 periods, however, one-way ANOVA was performed, why did not use two-way ANOVA, please explain.
  9. Actually, it is unnecessary to mark the same character in the figure, such as figure 1.

Author Response

Reviewer 3

This manuscript evaluated effects of nitrite exposure on the hematological properties, antioxidant and stress response of juvenile hybrid groupers. The research is meaningful. However, results of the experiment were a bit poor. The mechanism of response to nitrite exposure needs to investigate in depth.

L117: NO2-/L should be NO2-/L.

: We revise the wrong expression as below.

2.1. Pre-acute nitrite exposure experiment

The lethal concentration for 50% (LC50) of hybrid grouper for nitrite exposure was 856.79 mg NO2-/L, and the 96h LC50 of hybrid grouper was demonstrated in Table 1.

L143: table 1 should be table 2.

: We revise the wrong expression as below.

2.2. Experimental fish

Juvenile hybrid grouper, E. lanceolatus â™‚ x  E. fuscoguttatus â™€ (mean weight: 36.8 ± 4.2 g, mean length: 13.1 ± 0.6 cm) were obtained from a local fish farm in Korea. Fish used in this experiment were kept in a lab environment for 3 weeks before the experiment, and the seawater components are as shown in the Table 2. 

Please provide feeding information before experiment.

: We add feeding information as your suggestion.

2.2. Experimental fish

Juvenile hybrid grouper, E. lanceolatus â™‚ x  E. fuscoguttatus â™€ (mean weight: 36.8 ± 4.2 g, mean length: 13.1 ± 0.6 cm) were obtained from a local fish farm in Korea. Fish used in this experiment were kept in a lab environment for 3 weeks with feeding twice a day for fish health status before the experiment, and the seawater components are as shown in the Table 2. 

L149: How to ensure the stable nitrite concentration of each tank when change water once every 2 days?

: After completely changing the water repeatedly 3 times, the same concentration was adjusted using a standard stock solution of nitrite.

There were 10 tanks for 5 concentrations and 2 periods, thus for one treatment only one tank? No replications?

: In this study, repeat groups according to concentration groups were not calculated. Six fish for each concentration were used for analysis. In future research, we will set up a repeating section of the tank to conduct the experiment as the reviewer said.

Please check font size in table 3.

: We revise the wrong font size as your comment.

Table 3. Analyzed waterborne nitrite concentration (NO2- mg/L) from each source.

Waterborne nitrite concentration (NO2- mg/L)

Waterborne nitrite levels

0

10

20

40

80

Measured nitrite levels

0.07

10.5

21.2

43.9

83.3

L181: 10,000 rpm was not too high?

: The analysis method was made refering various article, and we use the methods for a lot of research. If there is a more appropriate method, we will refer to it for future research.

The experiment was designed 2 periods, however, one-way ANOVA was performed, why did not use two-way ANOVA, please explain.

: As the reviewer said, if use two-way ANOVA, it will be able to evaluate the effect of time and concentration. However, in this study, we focused on the toxic effects of exposure concentrations. In a future study, we will also conduct a study that reflects the exposure period together using two-way ANOVA according to the opinion of the reviewers.

We sincerely thanks for your careful review to improve this article.

Round 2

Reviewer 2 Report

it can be accepted.

Reviewer 3 Report

This manuscript could be accepted now.